# Association of variations in HLA class II and other loci with susceptibility to *EGFR*-mutated lung adenocarcinoma

Kouya Shiraishi[1], Yukinori Okada[2,3,4], Atsushi Takahashi[4,5], Yoichiro Kamatani[4], Yukihide Momozawa[6], Kyota Ashikawa[6], Hideo Kunitoh[7], Shingo Matsumoto[8], Atsushi Takano[9,10], Kimihiro Shimizu[11], Akiteru Goto[12], Koji Tsuta[13,14], Shun-ichi Watanabe[15], Yuichiro Ohe[16], Yukio Watanabe[15], Yasushi Goto[16], Hiroshi Nokihara[16], Koh Furuta[17,18], Akihiko Yoshida[13], Koichi Goto[19], Tomoyuki Hishida[20], Masahiro Tsuboi[20], Katsuya Tsuchihara[8], Yohei Miyagi[21], Haruhiko Nakayama[22], Tomoyuki Yokose[23], Kazumi Tanaka[11], Toshiteru Nagashima[11], Yoichi Ohtaki[11], Daichi Maeda[12], Kazuhiro Imai[24], Yoshihiro Minamiya[24], Hiromi Sakamoto[25], Akira Saito[26], Yoko Shimada[1], Kuniko Sunami[1,17], Motonobu Saito[1], Johji Inazawa[27,28,29], Yusuke Nakamura[30], Teruhiko Yoshida[25], Jun Yokota[31], Fumihiko Matsuda[32], Keitaro Matsuo[33], Yataro Daigo[9,10], Michiaki Kubo[6] & Takashi Kohno[1]

Lung adenocarcinoma driven by somatic *EGFR* mutations is more prevalent in East Asians (30–50%) than in European/Americans (10–20%). Here we investigate genetic factors underlying the risk of this disease by conducting a genome-wide association study, followed by two validation studies, in 3,173 Japanese patients with *EGFR* mutation-positive lung adenocarcinoma and 15,158 controls. Four loci, 5p15.33 (*TERT*), 6p21.3 (*BTNL2*), 3q28 (*TP63*) and 17q24.2 (*BPTF*), previously shown to be strongly associated with overall lung adenocarcinoma risk in East Asians, were re-discovered as loci associated with a higher susceptibility to *EGFR* mutation-positive lung adenocarcinoma. In addition, two additional loci, HLA class II at 6p21.32 (rs2179920; $P = 5.1 \times 10^{-17}$, per-allele OR = 1.36) and 6p21.1 (*FOXP4*) (rs2495239; $P = 3.9 \times 10^{-9}$, per-allele OR = 1.19) were newly identified as loci associated with *EGFR* mutation-positive lung adenocarcinoma. This study indicates that multiple genetic factors underlie the risk of lung adenocarcinomas with *EGFR* mutations.

[1] Division of Genome Biology, National Cancer Center Research Institute, Tokyo 104-0045, Japan. [2] Department of Human Genetics and Disease Diversity, Graduate School of Medical and Dental Sciences, Tokyo Medical and Dental University, Tokyo 113-8510, Japan. [3] Department of Statistical Genetics, Osaka University Graduate School of Medicine, Yokohama 230-0045, Japan. [4] Laboratory for Statistical Analysis, RIKEN Center for Integrative Medical Sciences, Yokohama 230-0045, Japan. [5] Omics Research Center, National Cerebral and Cardiovascular Center, Osaka 565-8565, Japan. [6] Laboratory for Genotyping Development, Center for Integrative Medical Sciences, RIKEN, Yokohama 113-8510, Japan. [7] Department of Medical Oncology, Japanese Red Cross Medical Center, Tokyo 150-0012, Japan. [8] Division of Translational Research, Exploratory Oncology Research and Clinical Trial Center (EPOC), National Cancer Center Research Institute, Chiba 277-0882, Japan. [9] Center for Antibody and Vaccine Therapy, Research Hospital, Institute of Medical Science, The University of Tokyo, Tokyo 108-0071, Japan. [10] Department of Medical Oncology and Cancer Center, Shiga University of Medical Science, Otsu 520-2121, Japan. [11] Department of Integrative Center of General Surgery, Gunma University Hospital, Gunma 371-8511, Japan. [12] Department of Cellular and Organ Pathology, Graduate School of Medicine, Akita University, Akita 010-8543, Japan. [13] Department of Pathology, National Cancer Center Hospital, Tokyo 104-0045, Japan. [14] Department of Pathology and Laboratory Medicine, Kansai Medical University, Osaka 573-1010, Japan. [15] Division of Thoracic Surgery, National Cancer Centre Hospital, Tokyo 104-0045, Japan. [16] Department of Thoracic Oncology, National Cancer Center Hospital, Tokyo 104-0045, Japan. [17] Department of Clinical Laboratories, National Cancer Center Hospital, Tokyo 104-0045, Japan. [18] Division of Clinical Laboratory, Kanagawa Cancer Center, Kanagawa 241-0815, Japan. [19] Department of Thoracic Oncology, National Cancer Center Hospital East, Chiba 277-0882, Japan. [20] Department of Thoracic Surgery, National Cancer Center Hospital East, Chiba 277-0882, Japan. [21] Molecular Pathology and Genetics Division, Kanagawa Cancer Center Research Institute, Kanagawa 241-0815, Japan. [22] Department of Thoracic Surgery, Kanagawa Cancer Center, Kanagawa 241-0815, Japan. [23] Department of Pathology, Kanagawa Cancer Center, Kanagawa 241-0815, Japan. [24] Department of Thoracic Surgery, Graduate School of Medicine, Akita University, Akita 010-8543, Japan. [25] Division of Genetics, National Cancer Center Research Institute, Tokyo 104-0045, Japan. [26] StaGen Co., Ltd., Tokyo 111-0051, Japan. [27] Department of Molecular Cytogenetics, Medical Research Institute, Tokyo Medical and Dental University, Tokyo 113-8510, Japan. [28] Graduate School of Medical and Dental Sciences, Tokyo Medical and Dental University, Tokyo 113-8510, Japan. [29] Bioresource Research Center, Tokyo Medical and Dental University, Tokyo 113-8510, Japan. [30] Department of Medicine and Department of Surgery, Center for Personalized Therapeutics, The University of Chicago, Chicago 60637, USA. [31] Genomics and Epigenomics of Cancer Prediction Program, Institute of Predictive and Personalized Medicine of Cancer (IMPPC), 08916 Badalona, Spain. [32] Center for Genomic Medicine, Graduate School of Medicine, Kyoto University, Kyoto 606-8501, Japan. [33] Division of Division of Molecular Medicine, Aichi Cancer Center Research Institute, Chikusa-ku, Nagoya 464-8681, Japan. Correspondence and requests for materials should be addressed to T.K. (email: tkkohno@ncc.go.jp).

Lung adenocarcinoma (LADC) is the most common type of lung cancer worldwide, with incidence and mortality rates increasing in both Asian and Western countries. Recent genome studies have subdivided LADC into several categories, with mutually exclusive activations of responsible driver onco-genes[1–3]. One subset of LADC is characterized by mutations in the gene encoding epidermal growth factor receptor (EGFR), a second with mutations in the Kirsten rat sarcoma viral oncogene homologue (KRAS) and a third with ALK, ROS1 or RET fusion, as well as other subcategories. EGFR mutations are present in 30–50% of LADCs in East Asian patients, a much higher frequency than in Caucasians (10–20%)[4,5]. A high proportion of patients with EGFR mutation-positive LADC are never-smokers and females, making the development of a preventive method critical[4,6–8]. Advanced LADCs with EGFR mutations are often inoperable and are treated with tyrosine kinase inhibitors; however, these tumours frequently become drug resistant, leading to disease progress and death[9]. Understanding the genetic factors underlying the development of LADC with EGFR mutation is required to elucidate disease aetiology and to identify effective methods of prevention.

Genome-wide association studies (GWASs) on lung cancer in populations of East Asian and European countries have found that several loci are associated with the risk of LDAC. These include loci at chromosomes 15q25.1 (CHRNAs: cholinergic receptor, nicotinic, alpha)[10–13], 5p15.33 (TERT: telomerase reverse transcriptase)[10,11,14–16], 3q28 (TP63: tumour protein p63)[10,14–16], 6p21.3 (BTNL2: butyrophilin-like 2)[14], 10q25.1 (VTI1A: vesicle transport through interaction with t-SNAREs 1A)[15] and 17q24.2 (BPTF: bromodomain PHD finger transcription factor)[14]. Although a TERT polymorphism was reported to be associated with the risk of non-small cell lung cancer with the EGFR mutation[17], inherited genetic factors underlying the risk for LADC with the EGFR mutation have not been comprehensively analysed. Here we performed a GWAS, followed by two validation studies, focusing on LADC with EGFR mutations.

## Results

### GWAS on the risk for LADC with EGFR mutations.
This study enrolled 6,867 LADC patients, all of whom were informative for EGFR mutation status by routine diagnosis or by the methods described in this study (Supplementary Table 1). Of the 6,867 patients, 3,173 (46.2%) were positive for EGFR mutations, a finding consistent with previous results in Japanese patients with LADC[5]. Some case and control subjects/data overlapped with those in our previous GWAS (Supplementary Table 2).

Germline DNAs of 663 EGFR mutation-positive LADC cases and 4,367 controls were genotyped using Illumina Omni1-Quad and OmniExpress chips, respectively (Supplementary Table 1). A quantile–quantile plot, generated using the results of a logistic regression trend test (Supplementary Fig. 1A), found that the genomic inflation factor ($\lambda_{GC}$) was 0.96. The lack of population substructure between these cases and controls was validated by principal component analysis (PCA) of the subjects (Supplementary Fig. 1B). Genotype clusters were visually inspected for the most strongly associated single-nucleotide polymorphisms (SNPs), and the results indicated a low possibility of false-positive associations resulting from genotype misclassification (Supplementary Fig. 2). In this GWAS, no locus reached genome-wide significance for association (that is, a logistic regression trend of $P < 5 \times 10^{-8}$) Supplementary Fig. 1C), including loci previously reported to be associated with overall LADC risk (Supplementary Table 2). Five SNPs in the four loci, TP63 at 3q28, TERT at 5p15.33, BTNL2 at 6p21.3 and BPTF at 17q24.3, were found to be more strongly associated with risk for LADC with EGFR mutation ($P_{Trend} < 10^{-4}$) than other SNPs identified in GWASs of European and Asian populations.

### Validation study.
To identify susceptibility loci, a validation study was conducted using two independent sample sets; the first validation cohort consisted of 1,275 cases and 6,817 controls, while the second validation cohort consisted of 1,235 cases and 3,974 controls (Supplementary Table 1). Of the 107 SNPs with $P < 1 \times 10^{-4}$ by a logistic regression trend test ($P_{Trend}$) in the GWAS, 43 were selected, with the other 64, located within the same locus ($r^2 > 0.8$), excluded. All 43 SNPs were genotyped successfully, using multiplex PCR-based Invader assays, in the germline DNAs of subjects in the first validation set. Ten SNPs showed odds ratios (Ors) in the same direction with $P_{Trend} < 0.05$ (Supplementary Table 3). These 10 SNPs were assayed in the germline DNAs of subjects in the second validation set (Supplementary Table 4). When the results of both validation sets were combined using a fixed-effects model, two SNPs, rs2179920 at 6p21.32 and rs2495239 at 6p21.1, showed significant associations after Bonferroni correction, in addition to five SNPs at four known LADC susceptibility loci (that is, $P_{Trend} < 1.1 \times 10^{-3}$ calculated as 0.05/43) (Table 1). When the results of the GWAS and the validation study were combined, both novel loci showed genome-wide significance (rs2179920 at 6p21.32; $P_{Trend} = 5.1 \times 10^{-17}$, OR = 1.36 and rs2495239 at 6p21.1; $P_{Trend} = 3.9 \times 10^{-9}$, OR = 1.19). ORs were similar between the GWAS and the validation studies with no heterogeneity. Thus, six loci, represented by seven SNPs, consisting of two at previously unidentified loci and four known loci described above, were associated with the risk of LADC with EGFR mutation (Table 1). There were no significant differences in the association of these seven SNPs with gender or smoking status, suggesting that these loci likely affected the risk for EGFR-positive LADC, irrespective of gender and smoking status (Supplementary Table 5).

### Differential association by EGFR mutation.
To assess the differential associations of these SNPs with LADC risk according to the presence/absence of EGFR mutations in tumour tissues, germline DNAs of 3,694 patients diagnosed with LADC without EGFR mutation were genotyped (Supplementary Table 1). Case–control analysis showed that all seven SNPs showed statistically significant or marginal association with risk for LADC without EGFR mutation, but with lower ORs than those for LADC with EGFR mutation (Supplementary Fig. 3 and Supplementary Table 6). Case–case analysis of tumours with and without EGFR mutations showed statistically significant allelic differentiation for four of the seven SNPs, rs2736100, rs3817963, rs2179920 and rs2495239, after Bonferroni correction (Supplementary Fig. 3 and Supplementary Table 7), indicating that these SNPs are significantly more strongly associated with the risk of LADC with than without EGFR mutation (that is, $P_{Trend} < 7.1 \times 10^{-3}$ calculated as 0.05/7).

### Imputation analysis of the HLA class II locus.
Imputation analyses were performed using the GWAS data for the two newly identified loci, rs2179920 at 6p21.32 and rs2495239 at 6p21.1, which were found to be more strongly associated with EGFR-positive than -negative LADC. The former SNP, rs2179920, was located in an intergenic region near the HLA-DPB1 (major histocompatibility complex, class II, DP beta 1) gene in the human leukocyte antigen (HLA) class II region, and was 630 kb proximal to rs3817963, a locus in the BTNL2 gene at the border between the HLA class II and class III regions. Therefore, imputation analysis was performed using the Japanese HLA imputation reference panel[18] that included the HLA class II

**Table 1 | Associations of seven SNPs with the risk of LADC with *EGFR* mutation.**

| SNP ID | Gene | Allele (risk allele) | Stage | Cases | | Controls | | *P*-value | OR (95% CI) | $P_{het}$ |
|---|---|---|---|---|---|---|---|---|---|---|
| | | | | Total | RAF | Total | RAF | | | |
| rs2736100 5p15.33 | *TERT* intron 2 | T/G (G) | GWAS* | 663 | 0.468 | 4,365 | 0.391 | $2.51 \times 10^{-7}$ | 1.37 (1.21–1.54) | |
| | | | First validation* | 1,274 | 0.480 | 6,816 | 0.385 | $8.98 \times 10^{-18}$ | 1.53 (1.38–1.69) | |
| | | | Second validation† | 1,221 | 0.461 | 3,974 | 0.386 | $3.64 \times 10^{-10}$ | 1.35 (1.23–1.48) | |
| | | | Combined validation studies‡ | 2,495 | 0.471 | 10,790 | 0.385 | $9.81 \times 10^{-26}$ | 1.43 (1.34–1.53) | 0.073 |
| | | | All studies combined‡ | 3,158 | 0.470 | 15,155 | 0.387 | $1.87 \times 10^{-31}$ | 1.42 (1.34–1.50) | 0.16 |
| rs2853677 5p15.33 | *TERT* intron 2 | T/C (C) | GWAS* | 663 | 0.377 | 4,367 | 0.308 | $2.96 \times 10^{-7}$ | 1.38 (1.22–1.56) | |
| | | | First validation* | 1,274 | 0.387 | 6,815 | 0.298 | $1.44 \times 10^{-16}$ | 1.53 (1.38–1.69) | |
| | | | Second validation† | 1,226 | 0.383 | 3,956 | 0.309 | $5.78 \times 10^{-11}$ | 1.38 (1.26–1.53) | |
| | | | Combined validation studies‡ | 2,500 | 0.370 | 10,771 | 0.302 | $1.47 \times 10^{-25}$ | 1.45 (1.35–1.56) | 0.17 |
| | | | All studies combined‡ | 3,163 | 0.383 | 15,138 | 0.303 | $3.32 \times 10^{-31}$ | 1.43 (1.35–1.52) | 0.31 |
| rs2179920 6p21.32 | *HLA-DPB1* intergenic | G/A (A) | GWAS* | 663 | 0.222 | 4,365 | 0.174 | $8.53 \times 10^{-5}$ | 1.34 (1.16–1.55) | |
| | | | First validation* | 1,272 | 0.204 | 6,817 | 0.154 | $2.12 \times 10^{-9}$ | 1.45 (1.28–1.63) | |
| | | | Second validation† | 1,229 | 0.214 | 3,971 | 0.174 | $5.56 \times 10^{-6}$ | 1.30 (1.16–1.46) | |
| | | | Combined validation studies‡ | 2,501 | 0.209 | 10,788 | 0.161 | $1.24 \times 10^{-13}$ | 1.37 (1.26–1.49) | 0.22 |
| | | | All studies combined‡ | 3,164 | 0.211 | 15,153 | 0.165 | $5.05 \times 10^{-17}$ | 1.36 (1.27–1.47) | 0.46 |
| rs3817963 6p21.3 | *BTNL2* intron 4 | A/G (G) | GWAS* | 663 | 0.388 | 4,365 | 0.327 | $2.33 \times 10^{-5}$ | 1.30 (1.15–1.48) | |
| | | | 1st validation* | 1,274 | 0.383 | 6,809 | 0.309 | $9.64 \times 10^{-11}$ | 1.38 (1.25–1.53) | |
| | | | Second validation† | 1,222 | 0.356 | 3,973 | 0.317 | $2.90 \times 10^{-4}$ | 1.20 (1.09–1.32) | |
| | | | Combined validation studies‡ | 2,496 | 0.370 | 10,782 | 0.312 | $1.05 \times 10^{-12}$ | 1.29 (1.20–1.38) | 0.041 |
| | | | All studies combined‡ | 3,159 | 0.374 | 15,147 | 0.316 | $1.20 \times 10^{-16}$ | 1.29 (1.21–1.37) | 0.12 |
| rs7636839 3q28 | *TP63* intron 1 | G/A (A) | GWAS* | 663 | 0.543 | 4,366 | 0.479 | $1.80 \times 10^{-5}$ | 1.29 (1.15–1.46) | |
| | | | First validation* | 1,274 | 0.520 | 6,813 | 0.478 | $9.22 \times 10^{-10}$ | 1.27 (1.18–1.37) | |
| | | | Second validation† | 1,232 | 0.530 | 3,972 | 0.480 | $1.62 \times 10^{-5}$ | 1.24 (1.12–1.38) | |
| | | | Combined validation studies‡ | 2,506 | 0.525 | 10,785 | 0.479 | $7.08 \times 10^{-9}$ | 1.21 (1.14–1.30) | 0.83 |
| | | | All studies combined‡ | 3,169 | 0.529 | 15,151 | 0.479 | $9.06 \times 10^{-13}$ | 1.23 (1.16–1.30) | 0.63 |
| rs7216064 17q24.3 | *BPTF* intron 9 | A/G (A) | GWAS* | 663 | 0.759 | 4,367 | 0.706 | $2.05 \times 10^{-5}$ | 1.35 (1.18–1.55) | |
| | | | First validation* | 1,275 | 0.741 | 6,817 | 0.708 | $5.54 \times 10^{-5}$ | 1.24 (1.12–1.38) | |
| | | | Second validation† | 1,231 | 0.749 | 3,974 | 0.703 | $4.34 \times 10^{-5}$ | 1.25 (1.12–1.39) | |
| | | | Combined validation studies‡ | 2,506 | 0.745 | 10,791 | 0.706 | $9.33 \times 10^{-9}$ | 1.24 (1.15–1.34) | 0.97 |
| | | | All studies combined‡ | 3,169 | 0.748 | 15,158 | 0.706 | $1.51 \times 10^{-12}$ | 1.27 (1.19–1.35) | 0.58 |
| rs2495239 6p21.1 | *FOXP4* intergenic | G/A (A) | GWAS* | 663 | 0.448 | 4,366 | 0.374 | $8.40 \times 10^{-7}$ | 1.34 (1.19–1.51) | |
| | | | First validation* | 1,273 | 0.413 | 6,817 | 0.383 | $5.21 \times 10^{-3}$ | 1.15 (1.04–1.26) | |
| | | | Second validation† | 1,231 | 0.422 | 3,966 | 0.387 | $4.70 \times 10^{-3}$ | 1.15 (1.04–1.26) | |
| | | | Combined validation studies‡ | 2,504 | 0.417 | 10,783 | 0.385 | $7.25 \times 10^{-5}$ | 1.15 (1.07–1.23) | 0.99 |
| | | | All studies combined‡ | 3,167 | 0.424 | 15,149 | 0.382 | $3.92 \times 10^{-9}$ | 1.19 (1.12–1.26) | 0.069 |

CI, confidence interval; RAF, risk allele frequency.
*Adjusted for age, gender and smoking status.
†Adjusted for gender and smoking status.
‡Combined meta-analysis was performed using a fixed-effects model.

genes, *HLA-DRB1*, *HLA-DQB1*, *HLA-DPA1* and *HLA-DPB1*, as described elsewhere. A case–control study of SNPs, including imputed ones, for *EGFR*-positive LADC risk revealed two association peaks (Fig. 1a), one including rs2179920 as a top 10 SNP and the other including rs3817963 as the top SNP (Supplementary Tables 8 and 9). The associations of both rs2179920 and rs3817963 SNPs with risk remained after conditioning on rs3817963 and rs2179920 genotypes, respectively (Fig. 1b,c and Supplementary Table 10). These results indicated that the HLA class II region contains two independent susceptibility loci. Interestingly, assessments of the

linkage disequilibrium of the risk allele of rs2179920 with non-synonymous SNP alleles of HLA class II genes showed that the strongest linkage disequilibrium for rs2179920 was with the Asp57 allele at the Glu57Asp SNP, which consists of multiple 4-digit *HLA-DPB1* alleles ($r^2 = 0.98$ in the binary analysis) (Supplementary Table 11). *HLA-DPB1* genotyping of 255 randomly chosen GWAS cases by the Luminex method validated that the imputation was highly accurate with a concordance rate of 99.1% for the Glu57Asp SNP between imputed and genotyped four-digit alleles. By contrast, rs3817963 of *BTNL2* did not show significant linkage disequilibriums

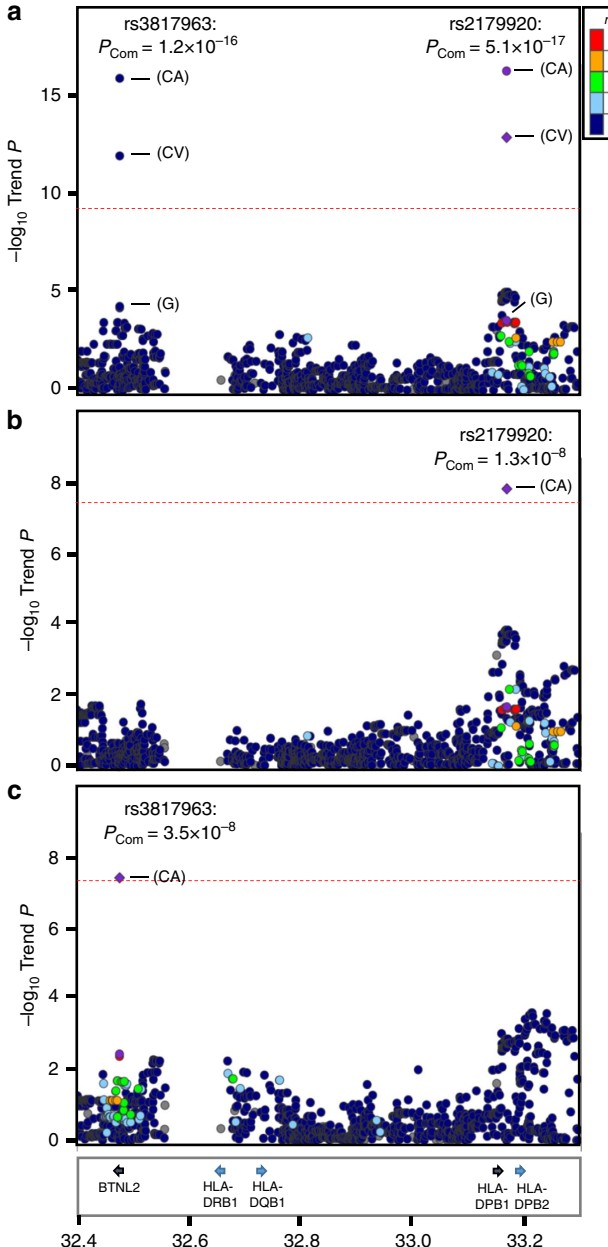

**Figure 1 | Regional plots of variations in the HLA class II region.** Plots show results of association studies of (**a**) nominal analysis, (**b**) conditioned analysis on rs3817963 and (**c**) conditioned analysis on rs2179920. Red line shows the level of genome-wide significance for association ($P_{Trend} < 5 \times 10^{-8}$). Genes within the region of interest are annotated and are indicated by arrows. The physical positions of the variants in the HLA class II region are shown at the bottom. Circles represent the location and $-\log_{10}$ ($P_{Trend}$ values) of each variant. The $-\log_{10} P_{Trend}$ values of the marker SNPs are shown for the GWAS (G), the combined validation study (CV) and the total combined study (CA).

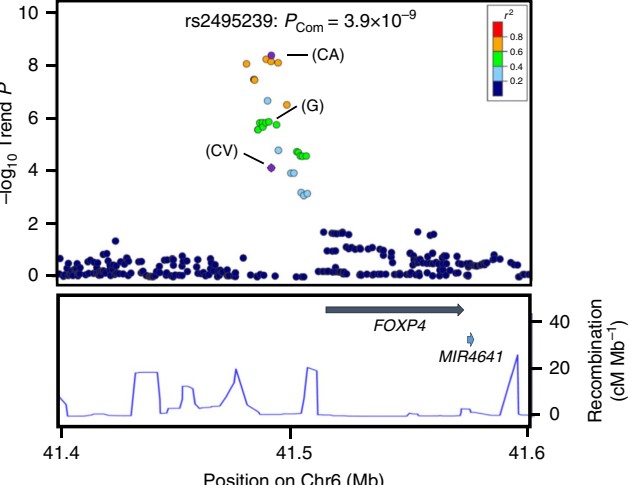

**Figure 2 | Regional plot of variations in the *FOXP4* region.** The marker SNP is shown in purple and the $r^2$ values for the other SNPs are indicated by different colours. Correlations were estimated using data from the 1000 Genomes Project. Genes within the region of interest are annotated and are indicated by arrows. The blue lines indicate the recombination rates in centimorgans (cM) per megabase (Mb). Circles represent the location and $-\log_{10}$ ($P_{Trend}$ values) of each variant. The $-\log_{10} P_{Trend}$ values of the marker SNPs are shown for the GWAS (G), the combined validation study (CV) and the total combined study (CA).

with non-synonymous SNPs of any HLA genes. In the GWAS cohort, the imputed variation at amino-acid residue 57 in *HLA-DPB1* showed a similar association with risk ($P_{Trend} = 6.0 \times 10^{-4}$) as rs2179920 ($P_{Trend} = 2.6 \times 10^{-4}$). On the other hand, none of the imputed *HLA-DPB1* alleles showed weaker associations than the marker SNP, rs2179920, and the Glu57Asp SNP (Supplementary Table 12). The amino-acid residue 57, which is located in one of the two extracellular domains of the HLA-DPB1 protein, does not make contact with antigen peptides[19,20]. Thus, the mechanism determining how the *HLA-DPB1* variation affects the risk of *EGFR* mutation-positive LADC should be further investigated, particularly with respect to the possibility that it affects immune responses against *EGFR*-positive tumour cells.

**Imputation analysis of the *FOXP4* locus.** Imputation analyses was also performed for the other newly identified SNP, rs2495239 at 6p21.1, using the EAS population panel from the 1000 Genomes Project database[21] (phase III). Twenty-four imputed and genotyped SNPs showed significant associations with risk of *EGFR*-positive LADC ($P_{Trend} < 10^{-4}$) (Fig. 2 and Supplementary Table 13). Four imputed SNPs showed genome-wide significance for association (Fig. 2). In fact, the rs7741164 SNP, the most strongly associated SNP, was also significantly associated with a higher susceptibility to *EGFR* mutation-positive LADC (Supplementary Table 14), indicating that 6p21.1 is an additional susceptibility locus. Interestingly, the rs2495239 SNP was located ~25 kb upstream of the *FOXP4* (forkhead box P4) gene, which encodes a member of the FOXP (forkhead transcription factor) family of proteins. FOXP proteins play key roles in cell cycle regulation and oncogenesis. Furthermore, meta-analysis, including those of studies in Japan and five other Asian countries, revealed that the *FOXP4* SNP, rs7741164, is associated with a higher risk of lung cancer risk among never-smoking Asian females[22], consistent with the aetiology of *EGFR* mutation-positive LADC[4]. Levels of *FOXP4* mRNA expression in 403 non-cancerous lung tissues, as determined by real-time quantitative PCR, differed according to the rs2495239 SNP ($P$ by the linear regression test = 0.023) genotype, suggesting high expression from the risk (A) allele (Supplementary Table 15). Therefore, a high level of *FOXP4* mRNA associated with a risk allele may increase the risk of LADC with *EGFR* mutation by promoting oncogenesis.

**Discussion**

This study has provided evidence for genetic susceptibility to the development of LADC with *EGFR* mutation by identifying six

susceptibility loci showing associations at genome-wide levels. Particularly, two new loci, 6p21.32 (*HLA-DPB1*) and 6p21.1 (*FOXP4*), and two known loci, 5p15.33 (*TERT*) and 6p21.3 (*BTNL2*), were revealed to be more preferentially associated with risk of *EGFR*-positive than *EGFR*-negative LADC. The re-discovery of the *TERT* and *BTNL2* loci is highly consistent with the fact that they these loci show strong associations with overall LADC risk, particularly in East Asians[10,11,14–16,22], that is, populations containing LADC cases with *EGFR* mutations. Further functional studies and genetic studies of other Asian and non-Asian populations are needed to clarify how these loci affect susceptibility to *EGFR* mutation-positive LADC. *EGFR*-positive LADC is more prevalent in East Asian than in European/American countries. Interestingly, the risk allele frequencies of several of the SNPs identified in this study, such as rs2495239 upstream of *FOXP4* (0.38 versus 0.08 in Supplementary Table 16), are different between Asians and Europeans. These differences may help elucidate the aetiology of LADC, including inter-ethnic variations.

## Methods

**Study design and subjects.** A three-stage GWAS of LADCs with *EGFR* mutations in the Japanese population was performed using independent samples. The characteristics of each case–control group are shown in Supplementary Table 1. The discovery GWAS samples consisted of 663 cases from the National Cancer Center Hospital (NCCH) and 4,367 controls from the BioBank Japan project[23] and the Osaka-Midosuji Rotary Club (MRC). The BioBank Japan project (see URLs), started in 2003, is a collaborative network of 66 hospitals throughout Japan that has collected genomic DNA, serum and clinical information from 300,000 patients to date diagnosed with any of 47 diseases to date. The MRC subjects were 988 healthy volunteers. Individuals with any cancer were excluded from the control group.

The validation study consisted of two independent cohorts. The first validation cohort included 1,275 patients with LADCs with *EGFR* mutations and 6,817 controls. The case subjects were 1,056 from the NCCH and 219 from Kanagawa Cancer Center. All control subjects were from the BioBank Japan project. Individuals with any cancer were excluded from the control group. The second validation cohort included 1,235 patients with LADCs with *EGFR* mutations and 3,974 controls (cancer-free volunteers) from the NCCH, Keio University, Tokyo and the Japan PGx Data Science Consortium (JPDSC)[24]. JPDSC subjects were obtained from 10 geographic regions in Japan. Of the case subjects, 900 were from the NCCH, 136 from Akita University Hospital and 199 from Gunma University Hospital. Of the control subjects, 2,823 were from JPDSC, 368 from NCCH and 783 from Keio University. Individuals with any cancer were excluded from the control group.

In addition, we conducted case–control studies consisting of 3,694 patients with LADCs without *EGFR* mutations and the same 15,158 controls described above. Case subjects included 3,148 from the NCCH, 117 from Kanagawa Cancer Center, 207 from Akita University Hospital and 224 from Gunma University Hospital.

Smoking history of cases and controls was obtained via interview using a questionnaire. Smokers were defined as those who had smoked regularly for 12 months or longer at any time in their life, whereas non-smokers were defined as those who had not. All LADCs were diagnosed by cytological and/or histological examination according to the WHO classification. NCCH case subjects in the GWAS and first validation sets consisted of cases having sufficient DNA for large-scale SNP analyses. Case subjects in the second validation set consisted of cases with smaller amounts of DNA, because, for example, only small quantities of non-cancerous tissues were available for DNA extraction. Genome-wide typing data of control subjects obtained before and after 2010 were used for the GWAS and the first validation study below, respectively. All the participants provided written informed consent. This project was approved by the ethics committees of all participating institutions, that is, BioBank Japan, the NCCH, JPDSC, Keio University, Kanagawa Cancer Center, Gunma University Hospital and Akita University Hospital.

**Sample preparation and genotyping.** Genomic DNA was extracted from peripheral blood leukocytes or non-cancerous lung tissues using QIAmp DNA Blood Maxi/QIAmp DNA Mini kits (Qiagen, Germany), according to the manufacturer's instructions.

In the GWAS, 678 cases of LADCs with *EGFR* mutations from the NCCH were genotyped using the Illumina HumanOmni1-Quad Chip. Controls in the GWAS consisted of 4,896 individuals with cerebral aneurysms, chronic obstructive pulmonary disease or glaucoma; these samples were genotyped using the Illumina Human OmniExpress Genotyping BeadChip. The genotype concordance rate between these two genotyping platforms has been shown to be >99% (ref. 10).

In the first validation cohort, 1,275 cases of LADC with *EGFR* mutation were genotyped using the multiplex PCR-based Invader assay (Third Wave

Technologies), as described[16]. Controls consisted of 6,817 individuals with epilepsy, nephrosis syndrome, atopic dermatitis, urinary tract stone disease or Basedow's disease; these DNA samples were genotyped using the Illumina Human OmniExpress Genotyping BeadChip. The same quality control criteria were applied as for GWAS (see below) to confirm that none of the control subjects in the first validation set was an unexpected duplicate or probable relative of a control subject in the GWAS. SNPs subjected to the first validation test were selected by using a cutoff $P_{Trend}$ value $< 1 \times 10^{-4}$, because associations of SNPs with ORs >1.3 and minor allele frequencies >0.3 were predicted to be detected with a statistical power >0.6.

In the second validation cohort, 1,235 cases of LADCs with *EGFR* mutations and 3,974 controls were genotyped using the TaqMan method or HumanOmni2.5 BeadChip kit, according to the manufacturer's protocol.

Genotyping of four-digit *HLA-DPB1* alleles was performed on randomly chosen 255 GWAS LADC cases by using a WAKFlow HLA typing kit (Wakunaga Pharmaceutical, Tokyo, Japan) together with the Luminex Multi-Analyte Profiling system (ThermoFisher Scientific, Waltham, MA) according to the manufacturers' instructions.

**Quality control of GWAS data sets.** Standard quality control was performed on scans, excluding those of individuals with gender discrepancies, low call rates ($<99\%$) or extremely out of Hardy–Weinberg equilibrium (that is, $P < 1.0 \times 10^{-6}$), as previously described[14]. If any first-degree pairs of relatives were apparent, the control was removed from a case–control pair; otherwise, the individual with the lower call rate was excluded. In addition, PCA was performed on the genotype data of the samples and on European (CEU), African (YRI), and east Asian (Japanese (JPT) and Han Chinese (CHB)) individuals obtained from the Phase II HapMap database using smartpca[25]. PCA revealed no evident population substructure and identified outliers for exclusion. Most subjects fell into a known main cluster (Hondo) of the Japanese population. The remaining 5,030 subjects, consisting of 663 cases with *EGFR* mutations and 4,367 control subjects, were used for GWAS. We also selected 631 cases without *EGFR* mutations based on the same criteria (Supplementary Table 1).

**Detection of *EGFR* mutations.** DNA samples from tumour tissues were screened for somatic mutations in *EGFR* exons 19 and 21 by high-resolution melting (HRM) analysis[26], invader assay[27], SCORPION-ARMS[28] or the PNA PCR clamp method[29] in this study[30]. *EGFR* mutation tests were carried out by several methods: PCR-Invader assay by BML Inc. (Tokyo, Japan); PNA-LNA PCR clamp assay by Mitsubishi Chemical Medience Corp. (Tokyo, Japan); Scorpion ARMS assay by SRL Inc. (Tokyo, Japan); and HRM assay by ourselves. The HRM assay was carried out using primer set A for detection of *EGFR* mutation in exon 19 and primer set B for detection of *EGFR* mutation in exon 21. The sequences of the primer set A were 5′-AAAATTCCCGTCGCTATC-3′ (forward) and 5′-AAGCAGAAACTCACA TCG-3′ (reverse). The sequences of the primer set B were 5′-AGATCACAGATT TTGGGGC-3′ (forward) and 5′-ATTCTTTCTCTTCCGCAC-3′ (reverse). These primers were designed by the Primer 3 software[31], UCSC Genome Browser[32] and BLAT analyses[33]. Tumour samples were obtained from cytological specimens, frozen tumour specimens and formalin-fixed, paraffin-embedded tissues.

**Imputation.** We performed two separate imputation analyses; one was imputation of the HLA region using the Japanese specific reference panel[18] and the other was a regional imputation using EAS references from the 1000 Genomes Project (Phase III).

In the HLA imputation, 939 control subjects were excluded from the GWAS study because they had been used to construct the Japanese reference panel. The imputation included two- and four-digit classical HLA alleles and amino-acid polymorphisms of class I and class II HLA genes, as well as additional SNPs[18].

Next, imputed data at 6p21.1 were obtained using the 1000 Genomes Phase 3 EAS reference panel. Shapeit2 (version 2.778)[34] was used for pre-phasing and Minimac2 (stamped 2014.9.15)[35] was used for imputation. In this analysis, data of all control subjects from the GWAS study were used. The EAS reference panel was prepared as follows; data for East Asian samples (CDX, CHB, CHS, JPT and KHV) were extracted from the 1000 Genomes phase 3 v5 reference vcf file. Then, monomorphic or singleton sites, multiallelic sites and sites with $P$ values obtained by the Hardy–Weinberg equilibrium test $< 10^{-6}$ were excluded. After imputation, poorly imputed SNPs defined by RSQR$<0.70$ were further excluded. Logistic regression analysis was performed by mach2dat software[36] (version 1.0.24) using imputed dosage as an explanatory variable under the additive genetic model. LocusZoom was used to draw regional association plots[37]. HaploReg v4.1 (ref. 38) was used to examine whether putative functional SNPs are present in the susceptibility loci.

**Statistical analysis.** In GWAS and the validation study set 1, the strength of association between SNPs and *EGFR* mutation-positive and -negative LADC was measured as ORs adjusted for gender, age (continuous) and smoking status (never or ever smoker) using an unconditional logistic regression analysis. In validation study set 2, ORs adjusted for gender and smoking status (never or ever smoker) were calculated. Combined analysis was performed using a fixed-effects model. Heterogeneity among studies was examined using the Breslow–Day test. The R

statistical environment version 2.6.1 (ref. 39), the JMP version 11.0 or PLINK1.06 (ref. 40) was used for statistical analyses.

**Association study of the rs7741164 SNP.** Genotype status of the rs7741164 SNP was imputed on all GWAS cases and controls, and 2,823 of the validation set 2 controls (Supplementary Table 1). The RSQR values, indicating imputation quality, were >0.8. The remaining samples, consisting of 2,510 cases (all validation set 1 cases and set 2 cases) and 7,968 controls (all validation set 1 controls and the remaining 1,151 validation set 2 controls), were genotyped using the invader assay or TaqMan method. These data were subjected to an association analysis, the results of which are shown in Supplementary Table 14.

**Expression analysis of non-cancerous lung tissues.** In all, 403 non-cancerous lung tissue samples were obtained from LADC patients undergoing resection at the NCCH. Total RNA was extracted from grossly dissected, snap-frozen tissue samples using TRIzol (Invitrogen, Carlsbad, CA, USA), according to the manufacturer's instructions. cDNA was synthesized using the SuperScript III first-strand synthesis system for reverse-transcription (RT)–PCR (Invitrogen) with random hexamers. Quantitative real-time (RT)–PCR analysis for *FOXP4* was performed on duplicate samples using the ABI PRISM 7900HT (Applied Biosystems, Foster City, CA, USA) with TaqMan primer and probe sets (Assay ID: Hs01055269_m1) from Applied Biosystems. Relative expression levels for each gene were normalized to that of *GAPDH* (catalogue# 4326317E, Applied Biosystems) and digitized according to the mean of two values. Linear trends between expression levels of *FOXP4* mRNA and increases in the number of risk alleles were tested in a multivariate linear regression model using JMP version 11.0 software (SAS Institute Inc., Cary, NC). The variables used for adjustment in each test were age (continuous), gender (male or female) and smoking status (never versus ever). A *P* value <0.05 by a multivariate linear regression test was considered statistically significant.

**Data availability.** All data in the GWAS analyses are available in Integrative Disease Omics Database (https://gemdbj.ncc.go.jp/omics/docs/others.html), under accession code GWAS031.

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

## Acknowledgements

We thank all of the subjects for participating in the study. We are grateful to the members of the National Cancer Center Biobank, BioBank Japan and the Rotary Club of Osaka-Mid-osuji District 2660 Rotary International in Japan for supporting our study. We thank Tomomi Aoi, Yoko Odaka, Misuzu Okuyama, Hirohiko Totsuka, Suenori Chiku, Takayuki Honda, Takashi Nakaoku, Masataka Takenaka, Tomoaki Yoshizawa, Karin Yokozawa, Noriko Abe, Sachiko Miura, Chizu Kina, Takahisa Kawaguchi, Meiko Takahashi, Hayato Konno, Hidemi Ito, Seiichi Kakegawa, Jun Atsumi, Kai Obayashi, Seshiru Nakazawa, Yoshiaki Takase and Aya Kuchiba, and technical staffs of the Center for Genome Medicine, NCC, for providing technical/methodological assistance. We also thank Hiroshi Hirose and Ikuo Saito for DNA samples of control subjects; and the Japan Pharmacogenomics Data Science Consortium (JPDSC), which is composed of Astellas Pharma Inc., Otsuka Pharmaceutical Co., Ltd, Daiichi-Sankyo Co., Ltd, Taisho Pharmaceutical Co., Ltd, Takeda Pharmaceutical Co Ltd, and Mitsubishi Tanabe Pharma Corporation, and chaired by Toshiro Heya of Takeda Pharmaceutical Co Ltd, for kindly providing the data. This research was supported in part by the Practical Research for Innovative Cancer Control from Japan Agency for Medical Research and Development (AMED: 16ck0106096h0003); the Japan Society for the Promotion of Science (JSPS) KAKENHI grant numbers 15H05911, 15H05670, 15K14429, 22590516, 19390359; the Princess Takamatsu Cancer Research Fund; and by the National Cancer Center Research and Development Fund (26-A-8 and 26-A-1:

NCC Biobank). This work was also conducted as part of the BioBank Japan Project, supported by the Ministry of Education, Culture, Sports, Science, and Technology, Japan (MEXT), and AMED; and was partly supported by a Grant-in-Aid of MEXT for Scientific Research on Innovative Areas- Resource and technical support platforms for promoting research (Platform of Supporting Cohort Study and Biospecimen Analysis).

## Author contributions

Ko.S., M.K. and T.K. designed the study; Y.O., Y.K., A. Takahashi, Y.M., K.A., N.K. and A.S. analysed the GWAS and replication data; H.S., A.S., Y.S., K.S., M.S., Te.Y., J.Y., A. Takano and Ko.S. performed the genotyping for the GWAS and replication study; H.K., S.M., Ki.S., A.G., K.T., S.-i.W., Y.O., Y.W., Y.G., Hi.N., K.F., A.Y., K.G., T.H., M.T., Y.M., Ha.N., To.Y., Katsu. T., Kazu. T., T.N., Y.O., D.M., K.I. and Y.M. recruited subjects and participated in diagnostic evaluations; Ko.S and T.K. wrote the manuscript; M.K., F.M., K.M., Y.D., J.I., Y.N. and T.K. contributed to the overall GWAS design.

## Additional information

**Competing financial interests:** The authors declare no competing financial interests.

