## [Peer review file · Nature Communications]

Reviewers' Comments:

Reviewer #1 (Remarks to the Author)

The manuscript titled 'Association of variations in HLA-class II and other loci with susceptibility to lung adenocarcinoma with EGFR mutation' evaluated the association between SNPs and the risk of developing lung adenocarcinomas with EGFR mutations by conducting a genome-wide association study, followed by two validation studies, in 3,173 Japanese patients with lung adenocarcinomas with EGFR mutations and 15,158 controls. They identified two novel loci (HLA-class II rs2179920 and FOXP4 rs2495239) as being associated with risk of lung adenocarcinomas with EGFR mutations.

Overall, the abstract, introduction, presentation of results, and conclusions are appropriate and concise and the paper is well written. The findings are novel and will be of substantial interest both to the study of lung cancer and overall to the study of genetic susceptibility to cancer and other chronic diseases.

The methods are appropriate but the following points should be addressed.

- 1) In each phase of their study, there is a question about whether or not the healthy controls represent the exposure (SNP) distribution of the study base that gave rise to the adenocarcinoma cases. Also, lung cancer cases were drawn from hospitals whereas the controls were from the general population. Some evidence should be provided that the population that gave rise to the cases and controls is the same, or comparable.
- 2) Cases and controls were genotyped on different platforms. The comparability of the platforms should be discussed.
- 3) The source of data for the exposure variable "smoking habit" should be provided and the authors should make the case that the data are comparable.
- 4) The authors should include a 'Statistical analysis' section in their Methods section and include more details (e.g. adjusted variables, how these were chosen and categorized) in the statistical methods used.
- 5) The authors should justify their use of $P < 1 \times 10^{-4}$ as the significance cutoff for selecting SNPs from the validation study.

Reviewer #2 (Remarks to the Author)

This manuscript by Shiraishi et al. performed genome-wide association study, using biobank in Japan and examined genomic variations in patients with lung adenocarcinoma. Genomes of 3173 patients with EGFR mutations, 3694 lung adenocarcinoma without EGFR mutations and 15158 patients controls were used. The authors discovered two novel loci, HLA class II at 6p21.32 and 6p21.1 (FOXP4) were identified as more strongly associated with risk of lung adenocarcinoma with EGFR mutations. Two known loci in TERT and BTNL2, previously shown to be strongly associated with lung adenocarcinoma risk in East Asians were also identified again.

GWAS was used to discover SNPs that correlate to occurrence of EGFR mutant lung cancer, or East Asian lung cancer patients. Several loci was identified. This probably is the largest cohort of EGFR mutant patients GWAS study in the literature. The idea and methodology was not novel, however, the discovery suggesting that HLA class II and FOXP4 may be contributing to occurrence of this

tumor does suggest that immune response may be involved in the development of EGFR mutant lung adenocarcinoma.

This study was well performed with training and validation. However, this study suffered from the limitation of majority of GWAS studies, i.e. low ORs (e.g. the OR for 6p21.1 validation set was only 1.15). Low ORs will lead to low predictive value and thus clinically these findings will not be useful. Although the authors tried to link the findings to HLA class 2 variations and FOXP4 expression, the evidence that these two genes are involved with higher risk to develop EGFR mutation lung cancer obviously is not solid. I suggest that more studies on that aspect should be included for the completeness of this manuscript.

In addition, the authors should give the criteria how training and validation set patients were determined.

Response to reviewers:

Response to reviewer #1 (Remarks to the Author): Expert in lung cancer genetics

The manuscript titled 'Association of variations in HLA-class II and other loci with susceptibility to lung adenocarcinoma with EGFR mutation' evaluated the association between SNPs and the risk of developing lung adenocarcinomas with EGFR mutations by conducting a genome-wide association study, followed by two validation studies, in 3,173 Japanese patients with lung adenocarcinomas with EGFR mutations and 15,158 controls. They identified two novel loci (HLA-class II rs2179920 and FOXP4 rs2495239) as being associated with risk of lung adenocarcinomas with EGFR mutations. Overall, the abstract, introduction, presentation of results, and conclusions are appropriate and concise and the paper is well written. The findings are novel and will be of substantial interest both to the study of lung cancer and overall to the study of genetic susceptibility to cancer and other chronic diseases. The methods are appropriate but the following points should be addressed.

Answer:

We express our appreciation to the reviewer for his/her insightful comments on our paper. The comments have helped us significantly improve the paper.

Q1. In each phase of their study, there is a question about whether or not the healthy controls represent the exposure (SNP) distribution of the study base that gave rise to the adenocarcinoma cases. Also, lung cancer cases were drawn from hospitals whereas the controls were from the general population. Some evidence should be provided that the population that gave rise to the cases and controls is the same, or comparable.

Answer:

We agree with the reviewer's comment. Thus, we performed a Principal-components analysis (PCA) on the GWAS samples, and showed that the controls and cases had a similar genetic background and therefore, were comparable. This is consistent with the fact that most cases and controls were from the main island of Japan. We have mentioned the results in the text (Lines 108-110) and added the data as Supplementary Fig. 1-B.

2) Cases and controls were genotyped on different platforms. The comparability of the platforms should be discussed.

Answer:

We agree with the reviewer's comment. We compared the genotyping platforms, Illumina HumanOmni1-Quad and OmniExpress Genotyping BeadChips. The genotype concordance rate between the two platforms has been shown to be > 99% (Wang et al., Nat Genet. 2014; 46(7):736-741); therefore, we think that our case and control data are comparable. Here, to confirm the concordance, we made a cluster plot of seven SNPs at susceptibility loci identified in GWAS and validation studies (newly presented as Supplementary Fig. 2). The clustering data indicated that major homozygotes, heterozygotes, and minor homozygotes were clearly judged. In addition, HWE of 7 SNPs was shown to be appropriate (newly presented as Supplementary Table 6 and 7), indicating a low possibility of false-positive associations resulting from genotype misclassification.

We have discussed this issue in the main text (Lines 105-106 and 110-112) and added the results to the revised paper (Supplementary Fig. 2 and Supplementary Tables 6 and 7). Evidence of genotyping concordance between the Illumina HumanOmni1-Quad and OmniExpress Genotyping BeadChip platforms is indicated in the Supplementary Note (Lines 54-55). We appreciate the reviewer's suggestion because this comment helped us strengthen our paper.

3) The source of data for the exposure variable "smoking habit" should be provided and the authors should make the case that the data are comparable.

Answer:

Thank you for pointing it out. Smoking history of cases and controls was obtained via interviews using a questionnaire. Smokers were defined as those who had smoked regularly for 12 months or longer at any time in their life, whereas non-smokers were defined as those who had not. We have clarified this point in the Supplementary Note (Lines 30-33). Therefore, "smoking habit" in the text has been replaced with "smoking status".

4) The authors should include a 'Statistical analysis' section in their Methods section and include more details (e.g. adjusted variables, how these were chosen and categorized) in the statistical methods used.

Answer:

We thank the author for this comment. We have added information on the detailed methods to the section dealing with statistical analysis in the Supplementary Note (Lines 118-126).

5) The authors should justify their use of $P < 1 \times 10^{-4}$ as the significance cutoff for selecting SNPs from the validation study.

Answer:

Using the cut-off P-value, we aimed to identify common SNPs showing OR > 1.3. We have mentioned this point in the Supplementary Note (Lines 63-66). Thank you for

pointing out the lack of such important information.

Reviewer #2 (Remarks to the Author): Expert in EGFR mutations

This manuscript by Shiraishi et al. performed genome-wide association study, using biobank in Japan and examined genomic variations in patients with lung adenocarcinoma. Genomes of 3173 patients with EGFR mutations, 3694 lung adenocarcinoma without EGFR mutations and 15158 patients controls were used. The authors discovered two novel loci, HLA class II at 6p21.32 and 6p21.1 (FOXP4) were identified as more strongly associated with risk of lung adenocarcinoma with EGFR mutations. Two known loci in TERT and BTNL2, previously shown to be strongly associated with lung adenocarcinoma risk in East Asians were also identified again.

GWAS was used to discover SNPs that correlate to occurrence of EGFR mutant lung cancer, or East Asian lung cancer patients. Several loci were identified. This probably is the largest cohort of EGFR mutant patients GWAS study in the literature. The idea and methodology was not novel, however, the discovery suggesting that HLA class II and FOXP4 may be contributing to occurrence of this tumor does suggest that immune response may be involved in the development of EGFR mutant lung adenocarcinoma.

Answer:

We express our appreciation to the reviewer for his/her insightful comments on our paper. The comments have helped us significantly improve the paper.

This study was well performed with training and validation. However, this study suffered from the limitation of majority of GWAS studies, i.e. low ORs (e.g. the OR for 6p21.1 validation set was only 1.15). Low ORs will lead to low predictive value and thus clinically these findings will not be useful. Although the authors tried to link the findings to HLA class 2 variations and FOXP4 expression, the evidence that these two genes are involved with higher risk to develop EGFR mutation lung cancer obviously is not solid. I suggest that more studies on that aspect should be included for the completeness of this manuscript.

Answer:

We agree with the reviewer's idea. The evidence for associations between *HLA-DPB1* and *FOXP4* loci and susceptibility to LADC is less than that for other known loci,

although, as the reviewer pointed out, our data were obtained from appropriate discovery and validation studies.

Thus, according to the reviewer's suggestion, we did additional studies on both of these loci.

HLA-DPB1:

- (1) In the original manuscript, evidence for association with the *HLA-DPB1* Glu57Asp variant came only from imputed GWAS data. To obtain more solid evidence, we genotyped 255 GWAS case subjects for 4-digit *HLA-DPB1* alleles using a Luminex assay commonly used in the clinic. This resulted in a concordance rate of 99.1% for the Glu57Asp SNP between the imputed and genotyped data, validating that the Asp57 variant is a strong candidate. We have referred to the results in the text (lines 172-176) and the Supplementary Note (lines 70-73).

- (2) Based on the reliability of the HLA imputation above, we addressed whether specific 4-digit *HLA-DPB1* alleles show stronger associations with susceptibility to *EGFR* mutation-positive LADC than the Asp57 variation. However, all of the imputed *HLA-DPB1* alleles showed weaker associations than the marker SNP, rs2179920, and the Glu57Asp SNP. Thus, it is unlikely that a specific *HLA-DPB1* allele(s) including Asp57 is responsible for the susceptibility. We agree that the mechanism of how the *HLA-DPB1* variation affects susceptibility to *EGFR* mutation-positive LADC should be further investigated. We have referred to these results in the text (lines 179-186) and show the results as Supplementary Table 12.

FOXP4:

- (1) Association analysis of rs7741164, the most strongly associated SNP in the imputation analysis of GWAS subjects, on all GWAS and validation set subjects was newly performed. The results showed a significant association of the SNP with *EGFR* mutation-positive LADC (Supplementary Table 14). For this purpose, we imputed the genotype status of the rs7741164 SNP of the GWAS of 663 cases and 4,367 controls; and the validation set 2 of 2,823 controls. We also genotyped the remaining samples, consisting of 7,968 controls and 2,510 cases, using the invader assay or TaqMan method. We have explained this point in the text (lines 192-195) and in the Supplementary Note (lines 128-136).

- (2) A recent meta-analysis including in Japan and five other Asian countries revealed

that the *FOXP4* SNP, rs7741164, is associated with the lung cancer risk of never-smoking Asian females (Wang et al., Hum. Mol. Genet 2016), consistent with the etiology of *EGFR* mutation-positive LADC. Thus, we have explained this point in the text (lines 198-202) and cited this paper as ref 25.

Meta-analysis of genome-wide association studies identifies multiple lung cancer susceptibility loci in never-smoking Asian women. Hum Mol Genet. 2016, Feb 1;25(3):620-9. doi: 10.1093/hmg/ddv494.

We believe that the additional studies have strengthened the candidacy of the two loci. However, we understand that further functional studies and genetic studies of other Asian and non-Asian populations will be needed to conclude and clarify how these loci affect susceptibility to *EGFR* mutation-positive LADC. Thus, we toned down the part of the Abstract dealing with the description of the significance of these loci (lines 59-69, 176-177 and 216-218).

In addition, the authors should give the criteria how training and validation set patients were determined.

Answer:

There were no solid criteria for subject selection based on clinical data. Subjects were chosen mainly based on the time we could obtain samples/data. NCC case subjects in the GWAS and first validation sets consisted of cases having sufficient DNA for large scale single nucleotide polymorphism (SNP) analyses, while subjects in the second validation set consisted of cases with smaller amounts of DNA, because, for example, only small quantities of non-cancerous tissues were available for DNA extraction. Genome-wide typing data of control subjects obtained prior to and after 2010 were used for the GWAS and the 1st validation study below, respectively. We explained this point in the Supplementary Note (Lines 34-40). We appreciate the reviewer's suggestion because it has allowed to improve the intelligibility of the paper for readers.

Reviewers' Comments:

Reviewer #1 (Remarks to the Author)

The authors have responded well to both reviews and I feel that the revised manuscript is acceptable for publication.

Reviewer #2 (Remarks to the Author)

The authors have replied the questions properly. I have no more comments and suggest approval for publication.